# Cross-sectional Learning of Extremal Dependence among Financial Assets

**Xing Yan**
School of Data Science
City University of Hong Kong
yanxing128@gmail.com

**Qi Wu**[*]
School of Data Science
City University of Hong Kong
qiwu55@cityu.edu.hk

**Wen Zhang**
JD Digits
zhangwen.jd@gmail.com

## Abstract

We propose a novel probabilistic model to facilitate the learning of multivariate tail dependence of multiple financial assets. Our method allows one to construct from known random vectors, e.g., standard normal, sophisticated joint heavy-tailed random vectors featuring not only distinct marginal tail heaviness, but also flexible tail dependence structure. The novelty lies in that pairwise tail dependence between any two dimensions is modeled separately from their correlation, and can vary respectively according to its own parameter rather than the correlation parameter, which is an essential advantage over many commonly used methods such as multivariate $t$ or elliptical distribution. It is also intuitive to interpret, easy to track, and simple to sample comparing to the copula approach. We show its flexible tail dependence structure through simulation. Coupled with a GARCH model to eliminate serial dependence of each individual asset return series, we use this novel method to model and forecast multivariate conditional distribution of stock returns, and obtain notable performance improvements in multi-dimensional coverage tests. Besides, our empirical finding about the asymmetry of tails of the idiosyncratic component as well as the market component is interesting and worth to be well studied in the future.

## 1 Introduction

Extreme market movements and rare events, which we call tail risk, play a very important role in portfolio investments, institutional risk management, and financial regulation. For a single asset, there has been a large body of literature concluding that the asset return follows non-normal distribution with significant heavy tails. A further complication is that the joint multiple asset return often exhibits nonnegligible tail dependence, which implies a higher chance of extreme co-movements than in the joint normal case or the independence case. Realizing that much effort has been made on univariate heavy tail modeling, it is in great need of specially designed models for multivariate tail dependence.

To measure tail dependence, the most common definition of the down-tail dependence coefficient of two random variables $X$ and $Y$ is defined as (see [Frahm et al., 2005]):

$$\lambda_{X,Y}^D = \lim_{\tau \to 0^+} \mathbb{P}\{X < Q_X(\tau), Y < Q_Y(\tau)\}/\tau, \tag{1}$$

where $Q_X(\tau)$ and $Q_Y(\tau)$ are $\tau$-quantiles of marginal distributions of $X$ and $Y$ respectively. Similarly, the up-tail dependence coefficient is defined as: $\lambda_{X,Y}^U = \lim_{\tau \to 1^-} \mathbb{P}\{X > Q_X(\tau), Y > Q_Y(\tau)\}/(1 - \tau)$. The tail dependence coefficient measures the degree of extremal, not typical, co-movements between two random variables. It is beyond the usual correlation which is a measure of average dependence. The reason why we should need tail dependence modeling is that the correlation

---

[*]Corresponding author.

is limited in assessing extreme dependence risk during financial crisis [Poon et al., 2003]. Ignoring tail dependence will also incur the huge volume of mis-pricing of credit derivatives that may cause disasters [Peng and Kou, 2009]. Actually, researchers have designed a systemic risk indicator using tail dependence and predicted well crisis-period stock returns [Balla et al., 2014]. Thus it is of great importance to the regulatory agencies. Besides, tail dependence of multivariate data is a general problem. It appears not only in financial markets, but also in energy markets [Aderounmu and Wolff, 2014], climatic data [Schoelzel and Friederichs, 2008], and hydrological data [Poulin et al., 2007].

The correlation also affects tail dependence since it is an average measure. But if one can separate the effect of correlation, should tail dependencies be different for different pairs of assets? Before figuring out this problem, let us review some existing approaches for modeling tail dependence. First, two normally distributed random variables with non-zero correlation have no tail dependence (the up/down-tail dependence coefficient is 0). So multivariate normal distribution may not be appropriate for modeling financial markets. Successful attempts to alleviate this shortcoming include switching to non-Gaussian multivariate distributions such as $t$ and elliptical, as well as resorting to the copula approach for constructing non-trivial dependence structures.

Multivariate $t$ is a direct extension of univariate $t$-distribution. Its parameter, the degrees of freedom, represents tail heaviness of each individual dimension and controls pairwise tail dependence as well. Although it is non-zero, the tail dependence coefficient is the same for each pair of dimensions when removing the correlation, and for up and down sides (see [Chan and Li, 2008]). Elliptical distribution is a more general distribution family than multivariate $t$ is. It covers lots of known distribution families including multivariate $t$, Laplace, power exponential, Kotz distribution, etc. From [Lesniewski et al., 2016], the tail dependence coefficient of elliptical distribution is mathematically difficult to be expressed exactly. But like multivariate $t$, its pairwise tail dependence coefficients vary only according to pairwise correlations. There are no freedoms for tail dependencies themselves. Copula approach is flexible enough to describe many tail dependence structures theoretically (see introductions in [Demarta and McNeil, 2005], [Embrechts et al., 2001], and [Aas et al., 2009]). There has been some literature using copula to model tail dependence of financial returns, such as [Jondeau and Rockinger, 2006] and [Frahm et al., 2005]. But most of them only deal with two markets or two assets. This is due to the mathematical complexity of copula when modeling three or more variables. Some other approaches model tail dependence of assets using non-parametric method or quantile regression, see [Poon et al., 2003] and [Beine et al., 2010].

So far, no flexible but compact tail dependence model for multiple assets exists. Noticing that joint extreme events come from not only joint daily fluctuation of asset prices but also some tail shock that has widespread impacts (e.g., the collapse of Lehman Brothers), we realize that it is quite necessary to model tail dependence separately from the correlation. This is also more realistic if we look at assets having different financial fundamentals, or coming from different sectors or asset classes. While their pairwise correlations are definitely different, their pairwise tail dependencies should be different too.

In this paper, we solve the problem by proposing a transformation of some known random vector. The resulting new random vector has distinct marginal tail heaviness. More importantly, the novelty lies in that pairwise tail dependence of any two dimensions can vary according to its own parameter rather than the correlation parameter. So tail dependence is modeled separately from the correlation. We show its flexible tail dependence structure through simulation. We also propose an intuitive learning algorithm to fit the novel model with data. Coupled with a GARCH filter to describe the conditional multivariate distribution of asset returns, we evaluate our model by doing multi-dimensional coverage tests on the forecasts of conditional distribution and achieve better performances than all the competitors do.

## 2 Lower-triangular Tail Dependence Model

In [Yan et al., 2018], researchers proposed a novel parametric quantile function to represent a univariate distribution with asymmetrical heavy tails, and used it to model the time-varying conditional distribution of single asset returns. It is a monotonically increasing nonlinear transformation of the quantile function of some known distribution $Z_\tau$, $\tau \in (0, 1)$:

$$Q(\tau|\mu, \sigma, u, v) = \mu + \sigma Z_\tau \left( \frac{e^{uZ_\tau}}{A} + 1 \right) \left( \frac{e^{-vZ_\tau}}{A} + 1 \right), \tag{2}$$

where $Z_\tau$ can come from standard normal, $t$-distribution, or some other distribution. The resulting parametric function $Q(\tau|\mu, \sigma, u, v)$ is a quantile function too, meaning its inverse function exists and is a distribution. $u \geq 0$ and $v \geq 0$ control its right and left-tail heaviness respectively. If in the usual case where $Z_\tau$ is standard normal, this new quantile function has an inverted S-shaped Q-Q plot against standard normal so that it can represent heavy-tailed distribution. In this paper, we propose a simpler form:

$$Q(\tau|\mu, \sigma, u, v) = \mu + \sigma Z_\tau \left( \frac{u^{Z_\tau}}{A} + \frac{v^{-Z_\tau}}{A} + 1 \right) := \mu + \sigma g(Z_\tau|u, v). \qquad (3)$$

Now $u \geq 1$ and $v \geq 1$ are forced. $g$ is used for simplifying the notation. This produces very similar heavy tail effects and makes the tail heaviness less sensitive to $u$ and $v$ when they become large, which is good for numerical computing and analysis.

Then we argue that this is equivalent to making the same transformation to the corresponding known random variable $z$: $y = \mu + \sigma g(z|u, v)$. The new random variable $y$ has quantile function being exactly Equation (3), because of the following lemma:

**Lemma 1** *If $X$ is a continuous random variable and has continuous quantile function $Q_X(\tau)$, $\tau \in (0, 1)$. $Y$ is a function of $X$: $Y = f(X)$, where $f$ is continuous and strictly increasing. Then $Y$ has quantile function $Q_Y(\tau) = f(Q_X(\tau))$.*

This inspires us to extend the univariate heavy-tailed quantile function in Equation (3) to the multivariate case by transforming random variables. Recall that a set of i.i.d. standard normal random variables $\boldsymbol{z} = [z_1, \ldots, z_n]^\top$ can compose any multivariate normal random vector by linear combination: $\boldsymbol{\mu} + B\boldsymbol{z}$, where $\boldsymbol{\mu}$ is the mean vector and one can restrict $B$ to be a lower triangular matrix with strictly positive diagonal entries. We make a direct extension to this and propose a new random vector $\boldsymbol{y} = [y_1, \ldots, y_n]^\top$ with individual heavy tails and pairwise tail dependencies by transforming the latent i.i.d. random variables $\boldsymbol{z} = [z_1, \ldots, z_n]^\top$ ($z_i$ can follow standard normal, $t$-distribution, etc.):

$$
\begin{aligned}
y_1 &= \mu_1 + \sigma_{11} g(z_1|u_{11}, v_{11}), \\
y_2 &= \mu_2 + \sigma_{21} g(z_1|u_{21}, v_{21}) + \sigma_{22} g(z_2|u_{22}, v_{22}), \\
&\ldots \\
y_n &= \mu_n + \sigma_{n1} g(z_1|u_{n1}, v_{n1}) + \sigma_{n2} g(z_2|u_{n2}, v_{n2}) + \cdots + \sigma_{nn} g(z_n|u_{nn}, v_{nn}).
\end{aligned}
\qquad (4)
$$

Here $\sigma_{ii} > 0$, $u_{ij} \geq 1$ and $v_{ij} \geq 1$. $A$ is a positive constant satisfying $A \geq 3$ (see [Wu and Yan, 2019]). We set $A = 4$ in this paper. Now we have totally $n$ location parameters $\mu_1, \ldots, \mu_n$, $(n^2+n)/2$ usual correlation/scale parameters $\sigma_{11}, \sigma_{21}, \sigma_{22}, \ldots, \sigma_{nn}$, $(n^2+n)/2$ right-tail parameters $u_{11}, u_{21}, u_{22}, \ldots, u_{nn}$, and $(n^2 + n)/2$ left-tail parameters $v_{11}, v_{21}, v_{22}, \ldots, v_{nn}$. The total number of parameters is $n + 3(n^2 + n)/2$.

Our transformation is analogous to $\boldsymbol{\mu} + B\boldsymbol{z}$ but we add different tail heaviness for every $z_j$ in every $y_i$, i.e., $z_j$ is replaced by $g(z_j|u_{ij}, v_{ij})$ in the equation of $y_i$ ($j \leq i$). Because $y_1$ and $y_2$ both have the latent variable $z_1$, they are correlated and have tail dependence as well. Intuitively, the new random vector $\boldsymbol{y}$ has marginally different tail heaviness and distinct pairwise tail dependencies, which will be verified by us later. In addition, to make the model robust, sometimes we may want to reduce the number of parameters in Equation (4). One effective way to achieve this is to force $u_{11} = u_{i1}, v_{11} = v_{i1}, \forall i \geq 1$, $u_{22} = u_{i2}, v_{22} = v_{i2}, \forall i \geq 2$, and so on. Now the total number of parameters is reduced to $3n + (n^2 + n)/2$.

## 2.1 Pairwise Tail Dependencies

We have realized that it is challenging to obtain the exact tail dependence coefficients for pairs of dimensions of our proposed lower-triangular $\boldsymbol{y}$. So in this subsection, we qualitatively show that $\boldsymbol{y}$ has distinct pairwise tail dependencies. To reveal that, we numerically approximate the tail dependencies of $y_1$ & $y_2$ and $y_2$ & $y_3$, and analyze how they depend on model parameters. Noticing that the definition in Equation (1) is a limit, we approximate the down-tail dependence coefficient of $y_i$ & $y_j$ by choosing a very small $\tau$ and using $\lambda_{ij}^D(\tau) = \mathbb{P}\{y_i < Q_{y_i}(\tau), y_j < Q_{y_j}(\tau)\}/\tau$ as the proxy down-tail dependence. We set $\tau = 10^{-3}$, simulate $10^7$ observations of $y_i$ and $y_j$, and calculate the empirical value of $\lambda_{ij}^D(\tau)$. The latent $\boldsymbol{z}$ is standard normal in this analysis.

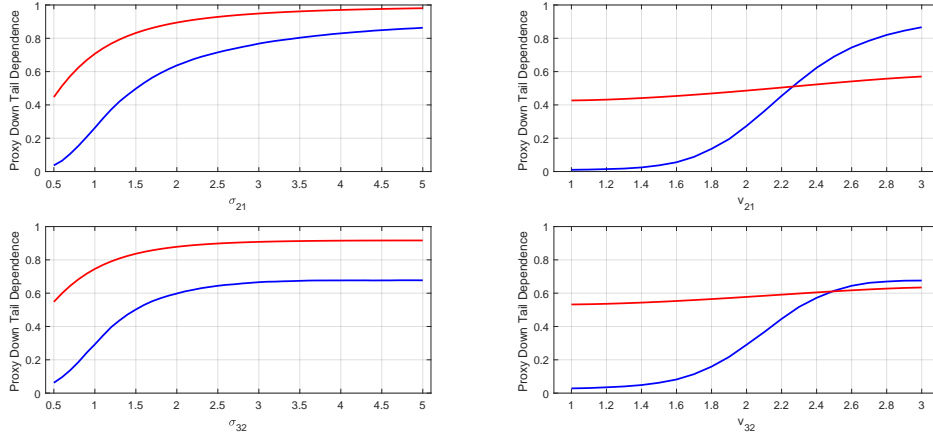

Figure 1: The plots of proxy down-tail dependencies (blue lines) and correlations (red lines) against varying model parameters (only one varies every time). The two subplots above are those of the pair $y_1$ & $y_2$ and the two blow are those of the pair $y_2$ & $y_3$. We change the parameters $\sigma_{21}$ and $v_{21}$ respectively for the pair $y_1$ & $y_2$ and change the parameters $\sigma_{32}$ and $v_{32}$ for the pair $y_2$ & $y_3$.

We first set $\mu_i = 0$, $\sigma_{ii} = 1$, $\sigma_{ij} = 0.5$ $(i > j)$, $u_{ii} = 1$, and $v_{ii} = 1.5$ for a 3-dimensional $\boldsymbol{y}$ of ours, and then change the value of one of these parameters every time to examine how the proxy down-tail dependencies $\lambda_{12}^D(\tau)$ and $\lambda_{23}^D(\tau)$ change accordingly. To distinguish tail dependence from usual correlation, we also examine how the usual correlation coefficients change accordingly. The results are shown in Figure 1. In the first subplot of Figure 1, when $\sigma_{21}$ varies, $\lambda_{12}^D(\tau)$ varies (blue line) in a much similar way as the usual correlation of $y_1$ & $y_2$ does (red line). This is a sign that the varying $\lambda_{12}^D(\tau)$ can be attributed to the effect of varying correlation. So $\sigma_{21}$ is the parameter that determines mainly the usual correlation. In the second subplot where $v_{21}$ varies, the proxy down-tail dependence $\lambda_{12}^D(\tau)$ changes nearly from 0 to 1 while the correlation stays nearly the same. This makes us conclude that $v_{21}$ is the dominant parameter determining the down-tail dependence of $y_1$ & $y_2$ separately from the correlation. The same analysis applies to the pair of $y_2$ & $y_3$, whose results are shown in the third and fourth subplots of Figure 1, where one can see $\sigma_{32}$ determines usual correlation and $v_{32}$ determines down-tail dependence of $y_2$ & $y_3$.

Our findings are consistent with our intuition about the tail dependence structure of $\boldsymbol{y}$. It can be extended that $v_{ij}$ is the parameter that determines mainly the down-tail dependence of $y_i$ & $y_j$. It also applies to the up-tail dependence situation, where $u_{ij}$ determines that of $y_i$ & $y_j$. Thus, our lower-triangular tail dependence model has a rather rich structure of tail dependence comparing to the commonly used multivariate normal, multivariate $t$, and elliptical distribution. We will also experimentally verify this in Section 4. Note that $v_{i1}, v_{i2}, \ldots, v_{ij}$ should all contribute to the down-tail dependence of $y_i$ & $y_j$ because they share the same latent $z_1, \ldots, z_j$. We believe $v_{ij}$ is the freest and dominant one that determines, because others contribute to that of $y_i$ & $y_{j-1}$ as well. Besides, we neglected an important situation in which the correlation parameter $\sigma_{ij}$ is negative. Negative correlation will lead to negative tail dependence. Different from the positive tail dependence we focus in this paper, it measures the dependence when one variable goes to positive extreme and the other goes to negative extreme, or vice versa, which is also common in financial markets. But fortunately, our above analysis also applies well and similar conclusions can be made.

## 2.2 One-factor Tail Dependence Model

Real world data is usually high-dimensional. In high-dimensional case, normally we need to simplify the model structure and reduce the number of parameters. This is also the idea behind factor analysis or principal components analysis. In finance, there are also factor models for asset returns. In this subsection, we propose a one-factor version of our tail dependence model for relatively high dimensional asset returns and we do not consider the asset pricing problem at this moment. Our one-factor model is a special case of the lower-triangular one in Equation (4).

**Algorithm 1** Algorithm for learning parameters of our proposed tail dependence model with data.

**Input**: $K$ observations $\{y_{:k}\}_{k=1}^{K}$. $y_{ik}$ is the $i$-th entry of the column vector $y_{:k}$.
**Parameters**: The positive constant $A \geq 3$ and the fine set of probability levels $\Psi \subset (0,1)$.
**Output**: All parameters $\mu_i, \sigma_{ij}, u_{ij}, v_{ij} (i \geq j)$ in our model.

1: **for** $i = 1, \ldots, n$ **do**
2:    **for** $j = 1, \ldots, i-1$ **do**
3:       Solve the following equation system to get the learned $\hat{\sigma}_{ij}, \hat{u}_{ij}, \hat{v}_{ij}$:

$$\frac{1}{K}\sum_{k=1}^{K} y_{ik}z_{jk}^{l} = \sigma_{ij}\frac{1}{K}\sum_{k=1}^{K} z_{jk}^{l}g(z_{jk}|u_{ij}, v_{ij}), \quad l = 1, 3, 5. \tag{6}$$

4:    **end for**
5:    Let $y'_{ik} = y_{ik} - \sum_{j=1}^{i-1} \hat{\sigma}_{ij}g(z_{jk}|\hat{u}_{ij}, \hat{v}_{ij})$, $k = 1, \ldots, K$. Solve the following quantile regression problem to get the learned $\hat{\mu}_i, \hat{\sigma}_{ii}, \hat{u}_{ii}, \hat{v}_{ii}$:

$$\min_{\mu_i,\sigma_{ii},u_{ii},v_{ii}} \frac{1}{K}\sum_{k=1}^{K}\sum_{\tau \in \Psi} L_\tau(y'_{ik}, Q(\tau|\mu_i, \sigma_{ii}, u_{ii}, v_{ii})). \tag{7}$$

6:    Solve the following equation to obtain realizations of $z_i$, i.e., $z_{i1}, \ldots, z_{iK}$:

$$y'_{ik} = \hat{\mu}_i + \hat{\sigma}_{ii}g(z_{ik}|\hat{u}_{ii}, \hat{v}_{ii}). \tag{8}$$

7: **end for**
8: **return** learned parameters $\hat{\mu}_i, \hat{\sigma}_{ij}, \hat{u}_{ij}, \hat{v}_{ij} (i \geq j)$.

For a market-wide or common variable $y_M$, and $n$ single-asset or individual variables $y_1, \ldots, y_n$, we model them as:

$$\begin{aligned} y_M &= \alpha_M + \beta_M g(z_M|u_M, v_M), \\ y_i &= \alpha_i + \beta_i g(z_M|u_i^M, v_i^M) + \gamma_i g(z_i|u_i, v_i), \quad i = 1, \ldots, n. \end{aligned} \tag{5}$$

$z_M, z_1, \ldots, z_n$ are latent i.i.d. random variables, e.g., standard normal. In financial context, $y_M$ can be the market return like S&P 500 (filtered by a GARCH-type model first, see our description later), and $u_M$ or $v_M$ represents its up or down heavy tail. $z_M$ is the market factor which is shared by every asset return $y_i$ (after filtering too). $y_i$ is decomposed into the market component and the idiosyncratic component. While $\beta_i$ is the average sensitivity of the $i$-th asset to the market factor, $u_i^M$ and $v_i^M$ can be seen as the tail-side sensitivities of the $i$-th asset. They cause all $y_i$ to have correlation-separated tail dependencies with $y_M$, as well as with each other. An extremal realization of $z_M$ will cause more additional impact on $y_i$ that cannot be captured by $\beta_i$ solely, which is an average-type sensitivity. And the tail dependence increases as $u_i^M$ or $v_i^M$ increases. The idiosyncratic component $\gamma_i g(z_i|u_i, v_i)$ of each asset is also heavy-tailed, and $u_i, v_i$ represent idiosyncratic heavy tails. In this paper, although we focus on financial modeling, this model can be applied to other fields too.

## 3 Parameter Learning

We propose a recursive-type learning algorithm to fit the proposed tail dependence model with data. This algorithm works for any choice of $\mathbf{z}$. It is a combination of quantile regression and method of moments. Because the one-factor version is a special case of the lower-triangular model, we only need to consider the learning algorithm for the lower-triangular model. The modifications we should make when applying to the one-factor version are straightforward. Suppose we have $K$ observations $\{y_{:k}\}_{k=1}^{K}$, where $y_{:k}$ is a column vector and $y_{ik}$ is its $i$-th entry. From the $y_1$ equation in Equation (4), we can conclude that $y_1$ has quantile function being in the form of Equation (3). So quantile regression [Koenker and Hallock, 2001] can be applied to learn the parameters of $y_1$ when a fine set of probability levels $\Psi \subset (0,1)$ is chosen:

$$\min_{\mu_1,\sigma_{11},u_{11},v_{11}} \frac{1}{K}\sum_{k=1}^{K}\sum_{\tau \in \Psi} L_\tau(y_{1k}, Q(\tau|\mu_1, \sigma_{11}, u_{11}, v_{11})). \tag{9}$$

Here $L_\tau$ is the loss function in quantile regression between the observation and $\tau$-quantile: $L_\tau(y, q) = (\tau - I(y < q))(y - q)$, where $I$ is indicator function. Please see [Yan et al., 2018] for an introduction of multi-quantile regression with a parametric quantile function. In our paper, we set $\Psi = \{0.01, 0.02, \ldots, 0.98, 0.99\}$ with 99 probability levels. Other smaller set that covers the interval $(0, 1)$ sufficiently is also acceptable, e.g., $\{0.01, 0.05, 0.1, \ldots, 0.9, 0.95, 0.99\}$ with 21 levels.

After solving the above optimization to get the learned parameters $\hat{\mu}_1, \hat{\sigma}_{11}, \hat{u}_{11}, \hat{v}_{11}$, one can inverse the $y_1$ equation in Equation (4) to obtain realizations of $z_1$ from $y_{1k}$. We denote them by $z_{11}, \ldots, z_{1K}$. Then, to learn the parameters of $y_2$, we multiply by $z_1$ on both sides of the $y_2$ equation in Equation (4) and take expectations. Noticing that $z_1$ and $z_2$ are independent and $E[z_1] = 0$, we have $E[y_2 z_1] = \sigma_{21} E[z_1 g(z_1|u_{21}, v_{21})]$. Replacing expectations by empirical averages leads to:

$$\frac{1}{K}\sum\nolimits_{k=1}^{K} y_{2k} z_{1k} = \sigma_{21} \frac{1}{K}\sum\nolimits_{k=1}^{K} z_{1k} g(z_{1k}|u_{21}, v_{21}). \tag{10}$$

This is one equation with three unknowns $\sigma_{21}, u_{21}, v_{21}$. If multiplying both sides by $z_1^3$ and $z_1^5$ instead, we obtain two more equations:

$$\frac{1}{K}\sum\nolimits_{k=1}^{K} y_{2k} z_{1k}^l = \sigma_{21} \frac{1}{K}\sum\nolimits_{k=1}^{K} z_{1k}^l g(z_{1k}|u_{21}, v_{21}), \quad l = 3, 5. \tag{11}$$

Solving the above three equations jointly gives us the learned parameters $\hat{\sigma}_{21}, \hat{u}_{21}, \hat{v}_{21}$.

After this, we consider a new random variable $y_2' = y_2 - \sigma_{21} g(z_1|u_{21}, v_{21}) = \mu_2 + \sigma_{22} g(z_2|u_{22}, v_{22})$ whose realizations are $y_{2k}' = y_{2k} - \hat{\sigma}_{21} g(z_{1k}|\hat{u}_{21}, \hat{v}_{21})$. And its quantile function is exactly in the form of Equation (3), or specifically, is $Q(\tau|\mu_2, \sigma_{22}, u_{22}, v_{22})$. So we can again apply quantile regression like in Equation (9) to learn $\mu_2, \sigma_{22}, u_{22}, v_{22}$. Actually, all the remaining parameters of $y_3, \ldots, y_n$ can be learned one by one following the same procedure. We summarize all the steps in Algorithm 1. Note that if one wants to reduce the number of parameters and restrict $u_{11} = u_{i1}$, $v_{11} = v_{i1}, \forall i \geq 1$, $u_{22} = u_{i2}$, $v_{22} = v_{i2}, \forall i \geq 2$, and so on, there will be only one equation and one unknown $\sigma_{ij}$ in Equation (6).

### 3.1 Modeling Multivariate Asset Returns

Suppose we have $n$ assets and their returns in $T$ days are $r_{it}$, $i = 1, \ldots, n$, $t = 1, \ldots, T$. To model the conditional distribution of $[r_{1t}, \ldots, r_{nt}]^\top$ using information up to time $t - 1$, we cannot ignore the serial dependence of each individual return series. The most recognized serial dependence of single asset returns is volatility clustering, which can be well captured by a GARCH-type model [Engle, 1982][Bollerslev, 1986]. We first adopt a AR(1)-GARCH(1,1)-like model to describe each asset return series individually:

$$\begin{aligned} r_t &= \mu_t + \sigma_t \varepsilon_t, \\ \mu_t &= \gamma_0 + \gamma_1 r_{t-1}, \\ \sigma_t^2 &= \beta_0 + \beta_1 (\sigma_{t-1} \varepsilon_{t-1})^2 + \beta_2 \sigma_{t-1}^2. \end{aligned} \tag{12}$$

For simplicity, we drop the subscript $i$ in the above equations. So for every time $t$, there are $n$ innovations $[\varepsilon_{1t}, \ldots, \varepsilon_{nt}]^\top$. We model them with our proposed tail dependence model, the lower-triangular or one-factor version, and suppose they are i.i.d. at time $t = 1, \ldots, T$.

The above model for multivariate asset returns is not easy to fit directly. So we take an indirect but effective way to do this. First, an AR(1)-GARCH(1,1) with $t$-distribution innovation is fitted to each return series. Then we collect all the residuals $[\hat{\varepsilon}_{1t}, \ldots, \hat{\varepsilon}_{nt}]^\top$, $t = 1, \ldots, T$ and fit our tail dependence model with them using Algorithm 1. For comparisons, other methods like multivariate normal, multivariate $t$, elliptical distribution, or copula approach can be used instead. We show the comparison results in Section 5.

## 4 Simulation Experiment

In this section, we experimentally verify the rich tail dependence structure of our model and compare it to the most widely used multivariate heavy-tailed distribution, the multivariate $t$-distribution, through simulation. On one hand, $10^6$ data points are sampled from a 3-dimensional $t$-distribution with 5 degrees of freedom and then we use this sampled data to fit our lower-triangular model with standard

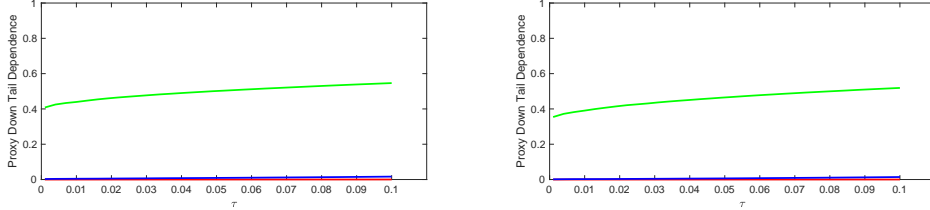

Figure 2: The proxy down-tail dependence $\lambda_{ij}^D(\tau)$ against $\tau$. The first subplot is from the 3-dimensional $t$-distribution we specify and the second one is from our model fitted using samples from the $t$-distribution. Three lines represent three pairs of dimensions.

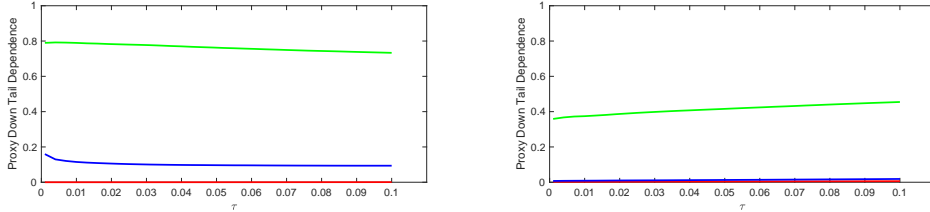

Figure 3: The proxy down-tail dependence $\lambda_{ij}^D(\tau)$ against $\tau$. The first subplot is from our lower-triangular model and the second one is from the 3-dimensional $t$-distribution fitted using samples of our model. Three lines represent three pairs of dimensions.

normal latent $\boldsymbol{z}$. Then proxy down-tail dependence $\lambda_{ij}^D(\tau) = \mathbb{P}\{y_i < Q_{y_i}(\tau), y_j < Q_{y_j}(\tau)\}/\tau$ for each pair of dimensions $i$ & $j$ is calculated for both the multivariate $t$-distribution and our model. We change $\tau$ in $[10^{-3}, 0.1]$ and plot $\lambda_{ij}^D(\tau)$ against the varying $\tau$ in Figure 2. The first subplot of Figure 2 is from the 3-dimensional $t$-distribution we specify. The second subplot is from our model fitted. Lines of different colors represent different pairs of dimensions. We see two models generate very similar line patterns, indicating that our model does capture the tail dependence structure of the $t$-distribution. Different levels of the 3 lines are due to different correlations of pairs.

On the other hand, inversely, $10^6$ samples from our lower-triangular model with standard normal latent $\boldsymbol{z}$ are generated and we fit a multivariate $t$-distribution using these samples. Again the proxy down-tail dependencies of every pairs of dimensions of these two models are calculated. We plot these $\lambda_{ij}^D(\tau)$ against $\tau$ in Figure 3, where we can see the 3-dimensional $t$-distribution (the second subplot) cannot generate line patterns that are close to those generated by our model (the first subplot), indicating that the $t$-distribution cannot capture the tail dependence structure of ours. This proves the flexible tail dependence structure of our model.

## 5  Conditional Distribution Forecasts

Our method described in Section 3.1 can forecast the conditional distribution of multiple asset returns on testing data after training. The training set and testing set are two successive multivariate time series of daily returns. We use statistical hypothesis testing to evaluate the forecasts. Recall that for evaluating univariate Value-at-Risk (VaR) forecasts, [Kupiec, 1995] proposed an unconditional coverage test that checks if the proportion of VaR violations in testing period is equal to the probability level of VaR. Inspired by this, we define a new type of violation for the two-dimensional case. Given random variables $(X, Y)$ and a fixed probability level $\tau$, suppose a $\tau^*$ solves the following equation:

$$\mathbb{P}\{X < Q_X(\tau^*), Y < Q_Y(\tau^*)\} = \tau, \tag{13}$$

where $Q_X(\tau^*)$ and $Q_Y(\tau^*)$ are $\tau^*$-quantiles of marginal distributions of $X$ and $Y$ respectively. If a realization of $(X, Y)$ is located in the area $[-\infty, Q_X(\tau^*)] \times [-\infty, Q_Y(\tau^*)]$, we say it is a violation, otherwise it is not. The probability of violation is obviously $\tau$. To solve Equation (13) to get $\tau^*$, we can use bisection method on many samples of $(X, Y)$ when the analytical distribution is not known.

Suppose for a sequence of pairs $\{(X_t, Y_t)\}$, we have forecasted its conditional distribution for every $t$. Given the realization of $\{(X_t, Y_t)\}$, i.e., the observations in the testing set, there is a sequence

Table 1: Unconditional coverage test statistic. (a) The first stock group: Apple, IBM, Microsoft. (b) The second: Apple, JP Morgan, Walmart. $\tau$ is 0.01 and $*$ represents the hypothesis is rejected at 95% confidence level. In parentheses it is the number of violations against ideal number of violations.

(a)

| Method\Pair | 1&2 | 1&3 | 2&3 |
|---|---|---|---|
| Normal | 6.96* (10/21) | 0.39 (18/21) | 0.83 (25/21) |
| $t$-distribution | 10.33* (8/21) | 2.50 (14/21) | 0.15 (19/21) |
| Clayton copula | 12.38* (7/21) | 3.37 (13/21) | 2.50 (14/21) |
| Gumbel copula | 0.83 (25/21) | 3.66 (30/21) | 15.54* (41/21) |
| Our model | 1.19 (16/21) | 0.24 (23/21) | 3.66 (30/21) |

(b)

| Method\Pair | 1&2 | 1&3 | 2&3 |
|---|---|---|---|
| Normal | 1.16 (19/24) | 8.98* (11/24) | 0.19 (22/24) |
| $t$-distribution | 2.34 (17/24) | 12.53* (9/24) | 0.41 (21/24) |
| Clayton copula | 2.34 (17/24) | 14.62* (8/24) | 2.34 (17/24) |
| Gumbel copula | 3.00 (33/24) | 3.99* (15/24) | 4.40* (35/24) |
| Our model | 0.15 (26/24) | 5.01* (14/24) | 0.15 (26/24) |

of whether the violation happens or not. Ideally, this 0-or-1 sequence should be samples from i.i.d. Bernoulli distribution with parameter $\tau$. To check if the proportion of violations in this sequence is $\tau$, Kupiec's test [Kupiec, 1995] for univariate case can be applied. The statistic of Kupiec's test is:

$$T_K = 2\log\left((1-\frac{m}{T})^{T-m}(\frac{m}{T})^m\right) - 2\log\left((1-\tau)^{T-m}\tau^m\right), \tag{14}$$

where $T$ is the length of the sequence, and $m$ is the number of violations. This statistic is asymptotically distributed on $[0, +\infty)$ as a chi-square with 1 degree of freedom. A zero of the statistic means the proportion of violations is exactly $\tau$. A large value of this statistic indicates the failure of forecasts. At 95% confidence level, the threshold for rejecting the hypothesis is 3.84. In our experiments, we set $\tau = 0.01$. For more than two assets, we do this test for any pair of assets while the model may be high-dimensional. From this, we can see if the pairwise tail dependencies are captured by the model. One needs to notice that the latent $z$ is always standard normal in the experiments.

## 5.1 Lower-triangular Model

We select two groups of stocks and evaluate our lower-triangular model as well as other competing methods on them. Each group contains 3 stocks that are representatives in the market. In the first group, 3 stocks from IT sector are selected: Apple, IBM, and Microsoft. In the second, 3 stocks are from different sectors: Apple, JP Morgan, and Walmart. The return data of these two groups start respectively from 14 March 1986 and 15 December 1980, and both end on 20 February 2019. They have 8,302 and 9,627 observations respectively. We leave the last one-fourth of the time series of each group as testing set. All returns are calculated by $r_t = 100\log(P_t/P_{t-1})$, where $P_t$ is the price.

For comparison, we also try competing methods including multivariate normal, multivariate $t$, Clayton copula [Clayton, 1978], and Gumbel copula [Kole et al., 2007]. The two copulas used here are bivariate. This is feasible when our evaluation is pairwise. We report the test statistic given by each method as well as the number of violations against the ideal number of violations in Table 1. In part (a) showing the results of the first stock group, we can see that while other methods all get at least one rejection at one of the three dimension pairs, our model performs without one rejection. It implies that our model does capture the distinct pairwise tail dependencies. In part (b) showing the results of the second stock group, our model performs fairly well on dimension pairs 1&2 and 2&3. The numbers of violations are very close to the ideal ones. Notice that all methods get rejected on dimension pair 1&3, suggesting the possibility of a regime-switching or similar thing happened from the training set to the testing set on that dimension pair. Overall, our proposed model does reach its purpose of design, as verified by the results shown here.

## 5.2 One-factor Model

To evaluate our one-factor model, data of 15 representative Dow-Jones stocks are collected such as AAPL, BA, JPM, and PG. The SP500 return serves as the market-wide variable. The multivariate return data starts from 1980-12-15 and ends at 2019-05-21 with 9,690 observations. Again the last one-fourth is left for testing. We use Algorithm 1 with very slight modification to fit our one-factor model with the multivariate data. The modification is easy to be obtained by the readers.

Table 2: The parameters of every asset in the one-factor model we have learnt (see Equation (5) for the introduction). SP500 is the market-wide variable and 15 representative stocks are selected into the model. The market variable SP500 has no parameters $\gamma_i, u_i, v_i$.

| Asset\Parameter | $\alpha_i$ | $\beta_i$ | $u_i^M$ | $v_i^M$ | $\gamma_i$ | $u_i$ | $v_i$ |
|---|---|---|---|---|---|---|---|
| SP500 | -0.0098 | 0.6814 | 1.6914 | 1.8241 | — | — | — |
| AAPL | 0.0256 | 0.3295 | 1.0000 | 1.7909 | 0.6071 | 2.0504 | 1.6992 |
| BA | -0.0223 | 0.3276 | 1.8424 | 1.8344 | 0.6043 | 1.7252 | 1.4940 |
| CAT | 0.0653 | 0.3882 | 1.0000 | 1.9623 | 0.5947 | 1.9417 | 1.6630 |
| CVX | 0.0368 | 0.3322 | 1.0000 | 1.9017 | 0.5898 | 1.7723 | 1.5853 |
| DIS | 0.0075 | 0.3557 | 1.6818 | 2.0383 | 0.5651 | 1.9214 | 1.6551 |
| DWDP | -0.0231 | 0.3644 | 1.7973 | 1.8581 | 0.5576 | 1.9340 | 1.6005 |
| IBM | 0.0686 | 0.4330 | 1.0000 | 1.9335 | 0.5216 | 2.0673 | 1.8113 |
| INTC | 0.0410 | 0.4240 | 1.0000 | 1.6414 | 0.5443 | 1.7529 | 1.5982 |
| JNJ | -0.0091 | 0.3430 | 1.8771 | 2.0107 | 0.5730 | 2.1325 | 1.7065 |
| JPM | 0.0581 | 0.4070 | 1.0000 | 2.0084 | 0.5393 | 1.9392 | 1.6606 |
| KO | 0.0219 | 0.3828 | 1.4783 | 1.9378 | 0.5759 | 1.9990 | 1.6679 |
| MMM | 0.0037 | 0.4023 | 1.8597 | 1.9330 | 0.5658 | 1.9396 | 1.7124 |
| NKE | -0.0083 | 0.3627 | 2.4380 | 1.9460 | 0.7101 | 3.1765 | 2.6941 |
| PG | 0.0335 | 0.3829 | 1.3750 | 1.9521 | 0.5998 | 2.0163 | 1.7129 |
| WMT | -0.0237 | 0.3769 | 1.8415 | 1.6061 | 0.5488 | 1.8287 | 1.6256 |

Our competing methods are one-factor Gaussian and one-factor $t$, in which we replace $g(z_M | \dots)$ and $g(z_i | \dots)$ in our one-factor Equation (5) by Gaussian/$t$-distributed $z_M$ and $z_i$. The degrees of freedom of the $t$-distribution can be different for different assets. Since we have 16 assets totally, there are $C_{16}^2 = 120$ pairs of dimensions. We evaluate these three models by reporting their numbers of rejections obtained in the 120 tests. Respectively, the one-factor Gaussian, $t$, and our model obtain 52, 43, and 32 rejections. The improvements are consistent with the intuition that when heavy tails are modeled and when the tails are separately modeled from correlations, the performances are better.

We have some interesting findings on the asymmetry of tails. In Table 2, we list the parameters of the 15 stocks as well as of SP500. It is not a coincidence when idiosyncratic components of all stocks are right-skewed, i.e., $u_i > v_i$ for all $i$. In contrast, for most stocks $u_i^M < v_i^M$, which means the market-wide tail impact is greater on the down side. A large SP500 drop will affect nearly all stocks severely while this is not the case on the up side. Actually, many stocks have $u_i^M = 1$, showing no tail sensitivity to the market variable on the up side. This deserves to be well studied in the future.

## 6   Conclusions

In summary, we propose a novel transformed random vector that is from some known random vector like standard normal, to learn the correlation-separated multivariate tail dependence structure of financial assets. We design it to let it have not only different marginal tail heaviness but also distinct pairwise tail dependencies. Our model has a lower-triangular version and a one-factor version. We also propose an algorithm to fit it with data. It is proved numerically to have distinct pairwise tail dependencies, which is an essential advantage over many commonly used methods. Combined with a GARCH-type model, we use it to forecast the conditional distribution of multi-dimensional asset returns and achieve significant performance improvements.

The empirical findings on the asymmetry of tails are interesting and worth to be well studied in the future. Many related questions need to be answered by further studies. For example, how to interpret the source of this idiosyncratic right skewness, what forms the market variable's left skewness when each component stock is right-skewed, and what their consequences are for asset pricing, either theoretically or empirically. Future works also include theoretical analysis on our model, especially the analytical tail dependence formula and the properties of the fitting algorithm.

## Acknowledgements

Qi WU acknowledges the financial support from the City University of Hong Kong grant SRG-Fd 7005300, and the Hong Kong Research Grants Council, particularly the Early Career Scheme

24200514 and the General Research Funds 14211316 and 14206117. This work was undertaken in part while Xing YAN was working with JD Digits.

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
