[Reviews · NeurIPS 2019]

Reviewer 1



-Did you run any experiments exploring the efficiency of the proposed fitting procedure? For a known model, how many samples are needed to reliably estimate the up/down tail dependence parameters? How does this vary with the strength of the dependence parameters (both rho and u/v)? -Can you provide more detail on the proposed fitting procedure. How many probability levels do you consider in the quantile regression? After estimating the parameters of y1, you invert g() to draw samples of z1, which are then used to estimate the parameters of y2, etc... It seems that the estimation errors will compound, making the estimate of the parameters of yn incredible noisy. Is it possible to remove some of the estimation error by proceeding along a different variable ordering? -Please define/address the constant A in eqn 2/3. -Why are u and v required to be greater than 1 in eqn 3? -How are the ideal number of violations in table 1 calculated?

Reviewer 2



The authors proposed a new statistical model for heavy tailed random vectors. They model the heavy tailed random vector as a known random vector transformed by a new quantile function. The model not only have parameters controlling the marginal tail heaviness, but also capture the tail dependence by separate parameters from the correlation parameters. To fit the model, the authors proposed an algorithm that recursively fits moment estimators and quantile regressions. The authors also combined the proposed model with a GARCH filter to describe the conditional distribution of financial asset returns, and demonstrated its superior performance by coverage tests in modeling historical stock return data. Quality: The technical content of the paper looks sound to me. The proposed model and fitting algorithm make intuitive sense, and the characterization of tail dependence is well demonstrated by simulation (Figure 1). Clarity: I find the methodology of the paper well presented with intuitive explanations. The experimental results are easy to understand as well in general, though a few implementation details could be revealed to improve clarity (see #3 and #4 below). Originality: The main contribution of the paper seems to be the generative model based on transformation by quantile functions, which captures correlations, tail heaviness, and tail dependence by separate parameters. The new quantile function seems novel, but supporting evidence justifying its advantage over existing ones seems to be missing (see #1 below). Coupling the GARCH-type models with heavy tailed models for the innovation terms is not new, though the experiment in Section 5.1 seems to show superior performance of the proposed model over common heavy tailed models. Significance: The paper makes relevant contribution on the methodology for modeling tail dependence. Based on the good simulation results, it looks promising for the methodology to be applied to modeling financial asset returns. On the other hand, one weakness of the paper is a lack of theoretical results on the proposed methodology. Most of the benefits of the new model have been demonstrated by simulations. It would be very helpful if the authors could provide some theoretical insights on the relation between the model parameters and the tail dependence measures, and on the performance (consistency, efficiency etc) of the parameter estimators. Itemized comments: 1. The advantage of the new quantile function (3) compared to the existing function (2) seems unjustified. Compared with (2), (3) changes the multiplicative factors containing the up and down tail parameters into an additive term. While this makes the function less sensitive to the tail parameters when they are large, the paper does not present supporting data on why the reduced sensitivity is desired. 2. On Line 132, the authors concluded that v_{ij} determines mainly the down-tail dependence of y_i and y_j. For any 1 <= k < j, does v_{ik} also have similar interpretation as v_{ij}? For example, in Equation (4), by symmetry, v_{31} and v_{32} seems to have similar effect on the tail dependence between y_3 and y_2. 3. In Algorithm 1 on Page 5, \Psi (the set of \tau's in Equation (7)) should also be an input parameter of the algorithm. Moreover, since it determines which quantiles are estimated in the loss function, I'd expect it to have notable effect on the results. I think it would be helpful to discuss how \Psi was chosen in the experiments, and provide some guidance on its choice in general. 4. Equation (13) doesn't seem to have closed form solution in general. Some details about how it's solved in the experiments and on the computational complexity would be helpful. 5. In addition to the up and down tail dependences, how could we also model negative tail dependence, e.g., P(X < Q_X(t), Y > Q_Y(1 - t)) / t? This is the counterpart of negative correlations, and is also notably common in financial asset returns (e.g., when money flow from one asset class (e.g., stocks) another (e.g., bonds)). Minor comments: 1. In Figures 2 and 3, it may be clearer to see the fitting errors if we overlay the oracle and the fitted lines in the same plot. Update: Thanks to the authors for the feedback. I believe Items 2 and 5 above are well addressed. On the other hand, as pointed out by another reviewer as well, a lack of theoretical results still seems to be the main weakness of the paper, though I agree that due to the complexity of the learning procedure, an extensive theoretical analysis would be a luxury at this stage.

Reviewer 3



The paper proposes a novel way to model correlations separately from tail dependence. The model comes in two flavors: lower-triangular model (an extension of the Gaussian vector factorization case) and one-factor tail dependence model. The authors demonstrate an algorithm to learn model parameters, and show through simulations its flexibility to model tail dependence and, combined with GARCH-type models, to forecast the conditional distribution of multiple asset returns. Originality I believe this is a novel approach to model tail dependency. Existing approaches such as the copula approach and elliptical distributions are compared. Quality The proposed model lacks theoretical analysis, e.g. its consistency properties but simulations are provided to ground the method. Note also that the model comes with great flexibility to model tail dependency but at the same time allows more degrees of freedom to overfit. I would be interested in a way to cope with this problem. Clarity The paper is quite well-written, symbols are clearly defined, and the algorithm is clearly specified. Significance This paper proposes a new way to address an important problem of tail dependence model with some evidence of method soundness

[Author Response · NeurIPS 2019]

**To Reviewer #1:**

**On the efficiency/consistency of the parameter learning procedure.** Please kindly see line 43-51.

**How many probability levels used?** All experiments use 99 levels of $\tau$ in the quantile regression: $\Psi=\{0.01,0.02,...,0.98,0.99\}$ so that the interval $(0,1)$ is covered sufficiently. One can use a smaller set to reduce the computing cost, e.g., $\Psi=\{0.01,0.05,0.1,...,0.9,0.95,0.99\}$. We find no significant performance difference using either the set of 99 levels or the 21 levels because the degree of freedom of the 4-parameter quantile function we use is 4. Both are sufficient.

**The error compounding issue.** The fitting of our one-factor model doesn't rely on variable ordering. Its estimation is quite reliable and efficient (line 43-50 has more details). For the lower-triangular model, more data would reduce the potential error compounding in $y_n$ as $n$ becomes large. We are trying to solve this issue by fine-tuning all parameters after the current learning procedure, with an overall objective instead of the step-by-step setting.

**The constant $A$ in Eqn.2 and Eqn.3.** We require $A$ be bounded below by a real constant such that Eqn.2 &3 are strictly increasing, which ensures Eqn.2 & 3 are indeed quantile functions. The constant $A$ determines the threshold starting from which our Q-Q plot starts to curve or say, be notably different from that of a Gaussian distribution.

**Why are $u \geq 1$ and $v \geq 1$ required in Eqn.3?** If $u < 1$, then $u^x = (1/u)^{-x}$ and $1/u > 1$ will play the same role as $v$ does, which is redundant. The same reason applies to $v$. We require the Q-Q plot exhibits both up and down tails.

**The ideal number of violations.** For the i.i.d. bernoulli distributed violation sequence with parameter $\tau$ (e.g., 0.01), the ideal number of violations is the number of observations $N$ times $\tau$. In Table 1, $N = 2075$ or $2407$, $\tau = 0.01$.

**Cross-sectional variance explained.** Our model is designed to capture tail dependence, which measures joint extreme events. They happened very rarely in markets, thus capturing them (fourth-order moment) will contribute little to the total variance explained (second-order moment). We have checked that our model does very slightly better than traditional one-factor statistical models in variance explained. Table 2 gives similar $\beta_i$ as CAPM, but with left/right tail sensitivity $u_i^M$ and $v_i^M$ added. The residual $\gamma_i g(z_i|u_i, v_i)$ is also heavy-tailed and asymmetric (described by $u_i$ and $v_i$).

**To Reviewer #2:**

**The advantage of Eqn.3 compared to Eqn.2.** We replace Eqn.2 by Eqn.3 for two reasons. First, in terms of controlling the shape of the left & right tail, the cross term $e^{(u-v)Z_\tau}/A^2$ in Eqn.2 is redundant and not "clean". The additive form of Eqn.3 avoids these undesirable properties. Second, changing from $e^{ux}$ to $u^x$ reduces the sensitivity of tail heaviness to $u$ while still allowing a wide range of tail heaviness. We found the $u^x$ form suits the experiments in Table 2 better.

**Does $v_{ik}$, $k < j$ have similar interpretation as $v_{ij}$?** Yes. For example, $v_{31}$ and $v_{32}$ both contribute to the tail dependence between $y_3$ and $y_2$. But from the fitting procedure, $v_{31}$ mainly determines tail dependence between $y_3$ and $y_1$, we believe $v_{32}$ is the most free parameter that determines tail dependence between $y_3$ and $y_2$. The full relationship between tail dependence and parameters is complicated. But in the one-factor model, it is clear and easy to interpret.

**How was $\Psi$ chosen in the experiments?** please kindly see line 3-5.

**How to solve Eqn.13?** The left side of Eqn.13 is an increasing function of $\tau^*$. We use the bisection method on $10^7$ samples of $(X, Y)$ to solve it. The computing cost is acceptable. We will introduce these in the next version.

**Could we also model negative tail dependence?** We appreciate the reviewer makes this point. We neglected to discuss the negative tail dependence $\lim_{t\to 0^+} \mathbb{P}\{X < Q_X(t), Y > Q_Y(1-t)\}/t$. Actually, our model does cover this case because it is symmetric, i.e., when $\sigma_{ij} < 0$, $i > j$, what we are modeling is exactly the negative correlation as well as negative tail dependence between $y_i$ and $y_j$. Now $u_{ij}$ and $v_{ij}$ are interpreted as the parameters controlling negative tail dependence.

**On the theoretical analysis of our model.** We share the same view that theoretical analysis of either the model or the estimator is important. We are actively developing it. For the consistency/efficiency matter of parameter learning, please kindly see 43-50.

**To Reviewer #3:**

**On the consistency/efficiency of the parameter learning procedure.** Developing a theory showing these properties is not easy, given the complexity of our learning procedure. But at this moment we can address this problem numerically. We randomly set the model parameters, simulate some points, and apply the learning procedure on them. The learned parameters of our one-factor model are shown in the following table ($i$ is any one in $1, 2, ..., n$). We also list the number of data points $N$ and the norm of learned parameters minus true ones. We can see the learning for one-factor model

| Trial\Parameter | $\alpha_M$ | $\beta_M$ | $u_M$ | $v_M$ | $\beta_i$ | $u_i^M$ | $v_i^M$ | $\alpha_i$ | $\gamma_i$ | $u_i$ | $v_i$ | $N$ | Norm |
|---|---|---|---|---|---|---|---|---|---|---|---|---|---|
| True Parameters | -0.33 | 0.60 | 2.08 | 2.34 | -0.71 | 1.84 | 1.83 | 0.88 | 0.18 | 2.37 | 2.12 | — | 0.00 |
| 1 | -0.33 | 0.56 | 2.36 | 2.67 | -0.64 | 1.75 | 2.57 | 0.82 | 0.23 | 1.84 | 2.26 | 1000 | 1.02 |
| 2 | -0.34 | 0.60 | 2.12 | 2.40 | -0.71 | 1.88 | 1.85 | 0.90 | 0.19 | 2.35 | 2.07 | 10000 | 0.10 |
| 3 | -0.33 | 0.60 | 2.10 | 2.36 | -0.71 | 1.85 | 1.85 | 0.88 | 0.19 | 2.36 | 2.10 | 100000 | 0.04 |

is fairly consistent and efficient as $N$ increases. For our lower-triangular model, the learning is less efficient but still consistent enough. The convergence becomes slower and we need more data points to obtain a reliable estimate. This is due to the more complex structure and more parameters of the model. For space limit we do not show its results here.

[Meta-Review · NeurIPS 2019]

This paper proposes a new approach to model tail dependence between random variables, i.e. dependence in the case of extreme events, which is different from naive correlation coefficient. Paper develops a methodology for estimating such models and presents a financial application. Experimental results are also provided. Overall, the results can be of interest beyond financial community and believe this would be a good contribution to Neurips this year.